# A denoising stacked autoencoders for transient electromagnetic signal denoising

Fanqiang Lin [1,4], Kecheng Chen [2,1], Xuben Wang [3,4], Hui Cao [3,4], Danlei Chen [1], and Fanzeng Chen [1]

[1]School of Information Science and Technology, Chengdu University of Technology, Chengdu,610059,China
[2]School of Computer Science and Engineering, University of Electronic Science and Technology of China, Chengdu,611731,China
[3]College of Geophysics, Chengdu University of Technology, Chengdu 610059,China
[4]Key Lab of Geo-Detection and Information Techniques of Ministry of Education, Chengdu 610059,China

**Correspondence:** Kecheng Chen(15775118240@163.com)

**Abstract.** Transient electromagnetic method (TEM) is extremely important in geophysics. However, the secondary field signal(SFS) in TEM received by coil is easily disturbed by random noise, sensor noise and man-made noise, which results in the difficulty in detecting deep geological information. To reduce the noise interference and detect deep geological information, we apply autoencoders,an unsupervised learning model in deep learning, on the basis of analyzing the characteristics of SFS, to denoise SFS. We introduce SFSDSA, a Secondary Field Signal Denoising Stacked Autoencoders,based on deep neural networks of feature extraction and denoising.SFSDSA maps the signal points of the noise interference to the high probability points with clean signal as reference according to the deep characteristics of the signal, so as to realize the signal denoising and reduce noise interference.The method is validated by the measured data comparison, and the comparison results show that the noise reduction method can effectively reduce the noise of SFS in contrast with the Kalman, PCA and wavelet transform methods, and strongly support the speculation of deeper underground features.

## 1 Introduction

Through the analysis of SFS in TEM, the information of underground geological composition can be obtained and has been widely used in mineral exploration, oil and gas exploration and other fields (Danielsen et al., 2003, Haroon et al., 2014).Due to the small amplitude of the late field signal in the secondary field , it may be disturbed by random noise, sensor noise, human noise and other interference(Rasmussen et al., 2017) which leads to data singularities or interference points, and thus the deep geological information can not be reflected well.Therefore, it is necessary to make full use of the characteristics of the secondary field signal to reduce the noise in the data and increase the effective range of the data.

Many methods have been developed for noise reduction of transient electromagnetic method. These methods can be broadly categorised into three groups: (1)Kalman filter algorithm(Ji et al., 2018),(2)Wavelet transform algorithm(Ji et al.,2016, Li et

al., 2017 ),(3)Principal component analysis(PCA) ( Wu et al.,2014).Kalman filtering is an effective method in linear systems, but it has little effect in nonlinear fields such as transient electromagnetic signals.The acquisition of wavelet threshold is cumbersome, and wavelet base selection is very difficult. In order to achieve the desired separation effect, to design an adaptive wavelet base is necessary. Likewise, the PCA algorithm is cumbersome too, some reseachers applied PCA to denoise the
trasient eleelectromagnetic signal, but the process of PCA requires at least 5 steps(Wu et al., 2014).

However,deep learning has been used to reduce noise from images, speech, and even gravitational waves(Jifara et al., 2017, Grais et al.,2017, Shen et al.,2017 ). Meanwhile,Autoencoder(AE)(Bengio et al., 2007),the representative model of deep learning,has been successfully applied in many fields (Hwang et al., 2016).AE with noise reduction capability(Denoising Autoencoders,DAE)(Vincent et al., 2008) has been widely used in image denoising (Zhao et al., 2014), audio noise reduction (Dai et
al., 2014), the reconstruction of holographic image denoising(Shimobaba et al., 2017) and other fields.

Nevertheless,in the field of geophysics, the application of deep learning model is limited (Chen et al., 2014).The use of deep learning model to reduce the noise of geophysical signals has not been applied.Therefore, in this paper, the Secondary Field Signal Denoising Stacked Autoencoders (SFSDSA) is proposed to reduce noise, based on a deep neural network with SFS feature extraction.SFSDSA will maps the signal points affected by noise to the high probability points with geophysical
inversion signal as reference according to the deep characteristics of the signal, so as to realize the signal denoising and reduce noise interference.

## 2   Related Work

Many works have bend done about the denosing of second field signal of transient electromagnetic method. Ji et al. proposed a method using wavelet threshold-exponential adaptive window width-fitting to denoise the second filed signal(Ji et al. 2016).
According to this method, stationary white noise and non-stationary electromagnetic noise can be filtered using the wavelet threshold-exponential adaptive window width-fitting to denoise the second filed signal, Li et al.used stationary-wavelet to denoise the electromagnetic noise in grounded electrical-source airborne transient electromagnetic signal(Li et al. 2017). This denoising algorithm can remove the electromagnetic noise from the grounded electrical source airborne transient electromagnetic signal. Wang et al. used wavelet-based baseline drift correction method for grounded electrical source airborne transient
electromagnetic signals, it can improve the signal-to-noise ratio(Wang et al. 2013). An exponential fitting-adaptive Kalman filter was uesed to remove mixed electromagnetic noises(Ji et al., 2017). It consists of an exponential fitting procedure and an adaptive scalar Kalman filter. The adaptive scalar Kalman uses the exponential fitting results in the weighting coefficients calculation.

The aforementioned kalman filter and wavelet transform are universal traditional filtering methods, which have their own
defects. However, the SFS itself has distribution characteristics, and the distortion of the waveform generated by the noise causes deviation from the signal point of the distribution.

The theoretical research indicates that (Bengio et al., 2007)the incomplete representation of autoencoders will be forced to capture the most prominent features of the training data and the high order feature of data is extracted,so autoencoders can

be applied to the feature extraction and abstract representation of SFS.Theoretical research also shows that (Vincent et al., 2008)), Denoising Autoencoders (DAE) can map the damaged data points to the estimated high probability points according to the data characteristics, to achieve the target of repairing the damaged data. Therefore, DAE can be appplied to map the SFS data points that will be disturbed by noise to the estimated high probability points, to achieve the purpose of SFS noise reduction.Studies have found (Vincent et al., 2010)the stacked DAEs (SDAE) have a strong feature extraction capability, and can improve the effect of feature extraction and enhance the ability of calibrating the deviation points disturbed by noise. SDAE is also commonly used in the compression encoding of the pre-processing height of complex images (Ali et al., 2017).

We also noticed that supervised learning performs well in classification problems such as image recognition and semantic understanding(He et al., 2016, Long et al., 2014). At the same time, unsupervised learning also has a good performance in clustering and association problems (Klampanos et al., 2018.), and the goal of unsupervised learning is usually to extract the distribution characteristics of the data in order to understand the deep features of the data (Becker and Plumbley, 1996, Liu et al., 2015). Both supervised learning and unsupervised learning have their own well-behaved areas, so we need to choose different learning styles and models for different problems. For the noise suppression problem of the SFS in TEM, our goal is to extract the deep features, and map the data points affected by noise to the estimated high probability points according to their own signal features. We also found that the purpose of extracting the distribution characteristics of the SFS data is similar to that of unsupervised learning. Meanwhile, unsupervised learning models are widely used in different signal noise reduction problems.

Therefore, based on the study of the distribution characteristics of the secondary field signal and autoencoders denoising method, we propose SFSDSA, a Secondary Field Signal Denoising Stacked Autoencoders, which is a deep learning model of transient electromagnetic signal denoising

(1)SFSDSA will be stacked by multiple AEs to form a deep neural network of multilayer owe complete coding, and multiple AEs are used as a higher-order feature extraction part, which can utilize its deep structure to maximize the characteristics of secondary field signal.

(2)Based on the principle of DAE, SFSDSA will set the secondary field measured data (received data)as the input data, and geophysical inversion method is used to process the measured data of the secondary field to obtain the inversion signal as the clean signal data. SFSDSA maps the signal points of the noise interference to the high probability points with clean signal as reference according to the deep characteristics of the signal. Because maintaining the original data dimension is especially important for the undistorted and post-processing of the signal, it is necessary to set the original dimension after the last coding as the output layer dimension. Although the output method may produce the decoding loss, it can have high abstract retention of the secondary field signal characteristics, and map the affected signal points to the high probability position points.

(3)The problem of too many nodes dying is a general disadvantage for RELU activation function and improved RELU activation functions like Leaky RELU all consistently outperform the RELU in some tasks(Xu et al., 2015). Therefore, it is necessary to apply the improved RELU function to reduce the impact of the shortcomings of the RELU function. We choose the SELU that have the preponderances of overcoming vanishing and exploding gradient problems in a sense and the best preforming in full connection networks(Klambauer et al., 2017). We chose Adam algorithm, which have the advantages

of calculating different adaptive learning rates for different parameters and requiring little memory(Kingma and Ba, 2014). Meanwhile, introducing regularized loss to solve the problems of over-fitting due try to increased depth and the SFSDSA only learning an identity function.

## 3 Mathematical Derivation of SFSDSA

Firstly, the secondary field data(actual detection signal)are treated as a noisy input. Since the secondary field data are mainly a time-amplitude value, we can sample the signal as a point-amplitude value, in the form of matrix $A$, the dimensions are $1 \times N$:

$$A = \begin{bmatrix} a_{11} & a_{12} & \cdots & a_{1n-1} & a_{1n} \end{bmatrix} \tag{1}$$

Secondly, the geophysical inversion method is used to obtain the theoretical signal, which can be used as a clean signal, then the theoretical signal is sampled as point-amplitude value, in the form of matrix $\tilde{A}$, the dimensions are $1 \times N$:

$$\tilde{A} = \begin{bmatrix} \tilde{a}_{11} & \tilde{a}_{12} & \cdots & \tilde{a}_{1n-1} & \tilde{a}_{1n} \end{bmatrix} \tag{2}$$

Thirdly, SFSDSA training model can be built, and Adam, which is a stochastic gradient descent (SGD) method, is applied to prevent gradient disappearance, and regularization loss is used to prevent over-fitting and SELU activation function is utilized to prevent too many points of death.

$$g_\theta(a_{1n}) = f_{SELU}(Wa_{ln} + b) \tag{3}$$

$$g_\theta(a_{1n}) = \lambda \begin{cases} Wa_{1n} + b & a_{1n} > 0 \\ \alpha e^{Wa_{1n}+b} - \alpha & a_{ln} <= 0 \end{cases} \tag{4}$$

Where $\theta = (w,b)$ , $w$ denotes the $N \times N'$ parameter matrix ($N' < N$), $b$ denotes the offset of the $N'$ dimensions. After the first compression coding layer, the signal is extracted features to $1 \times N'$.In order to extract high-level features while removing as much noise as possible and other factors, we can compress again.

$$g_{\theta'}(a'_{1N'}) = \lambda \begin{cases} W'a'_{1N'} + b' & a'_{1N'} > 0 \\ \alpha e^{W'a'_{1N'}+b'} - \alpha & a'_{1N'} <= 0 \end{cases} \tag{5}$$

$w$ denotes the $N' \times N''$ parameter matrix ($N'' < N'$), and $b$ denotes the offset of the $N''$ dimensions, and features of actual detection signal is extracted again, after more feature extraction layers can be stacked.For the secondary field signal, it is necessary to maintain the same input and output dimensions to ensure that the signal is not distorted and later processed. When feature extraction reaches to a certain extent, it is necessary to reconstruct back to input dimensions .

Reconstruction can be regarded as the process that the noisy signal points map back to the original dimensions after features being highly extracted .At the same time, reconstruction is the process of signal characteristic amplification. Finally output matrix $\bar{A}$ with the same dimensions as the inputs can be got :

$$\bar{A} = \begin{bmatrix} \bar{a}_{11} & \bar{a}_{12} & \cdots & \bar{a}_{1n-1} & \bar{a}_{1n} \end{bmatrix} \tag{6}$$

The output $\bar{A}$ we obtained can be used to get the loss from the clean signal $\tilde{A}$ using the loss function. The general loss function has square loss, which is mostly used in the linear regression problem. However, the secondary field data are mostly non-linear, and absolute loss is used in this paper:

$$L(\bar{A}, \tilde{A}) = |\bar{A} - \tilde{A}| \tag{7}$$

In the meantime, regularization loss optimization is used in this paper in order to avoid the problem of over-fitting, then:

$$loss = \theta^*, \theta'^* = arg_{\theta,\theta'} min \frac{1}{n} \sum_{i=1}^{n} L(x^i, g_{\theta'}(f_\theta(x^i))) + \lambda R(w) \tag{8}$$

After the loss is calculated, Adam algorithm is used to reverse optimization of parameters.

Figure 1 is the algorithm structure diagram of SFSDSA.With reference to the theory of DAE, SFSDSA maps the signal points of the noise interference to the high probability points with clean signal as reference according to the deep characteristics of the signal, so as to realize the signal noise and reduce noise interference. This high probability position is determined by the theoretical clean signal and the multi-layer model of the feature extraction ability.The multi-layer feature extraction makes the deep feature of secondary field data be preserved, and the effect of noise is reduced.

For the noise suppression problem of the secondary field signal in transient electromagnetic method, our goal is to extract the deep features of the secondary field signal, and map the data points affected by noise to the estimated high probability points according to their own signal features. We also found that the purpose of extracting the distribution characteristics of the secondary field signal data is similar to that of unsupervised learning.

## 4 Experiment and Analysis

In this paper, the secondary field signal of a certain place is used as the experimental analysis signal. Usually, the secondary field signals can be obtained continuously on a period of time, so a large number of signals can be extracted conveniently as the training samples.The secondary field actual signals are extracted as $1 \times 434$ as input signals of noise pollution, as is shown in the Figure 2(a).At the same time, based on the secondary field actual signals, the geophysical inversion method is used to obtain the theoretical detection signal as clean signal uncontaminated by noise, as is shown in the Figure 2(b).In order to be able to highlight the differences between the data, data are expressed in a double logarithmic form(loglog), as is shown in the Figure 3(a) and Figure 3(b).

The deep features of original data are abstracted by features extraction layers(compression coding layers). As the number of layers increases, SFSDSA can be a more complex abstract model with limited neural units, (to get higher-order features for

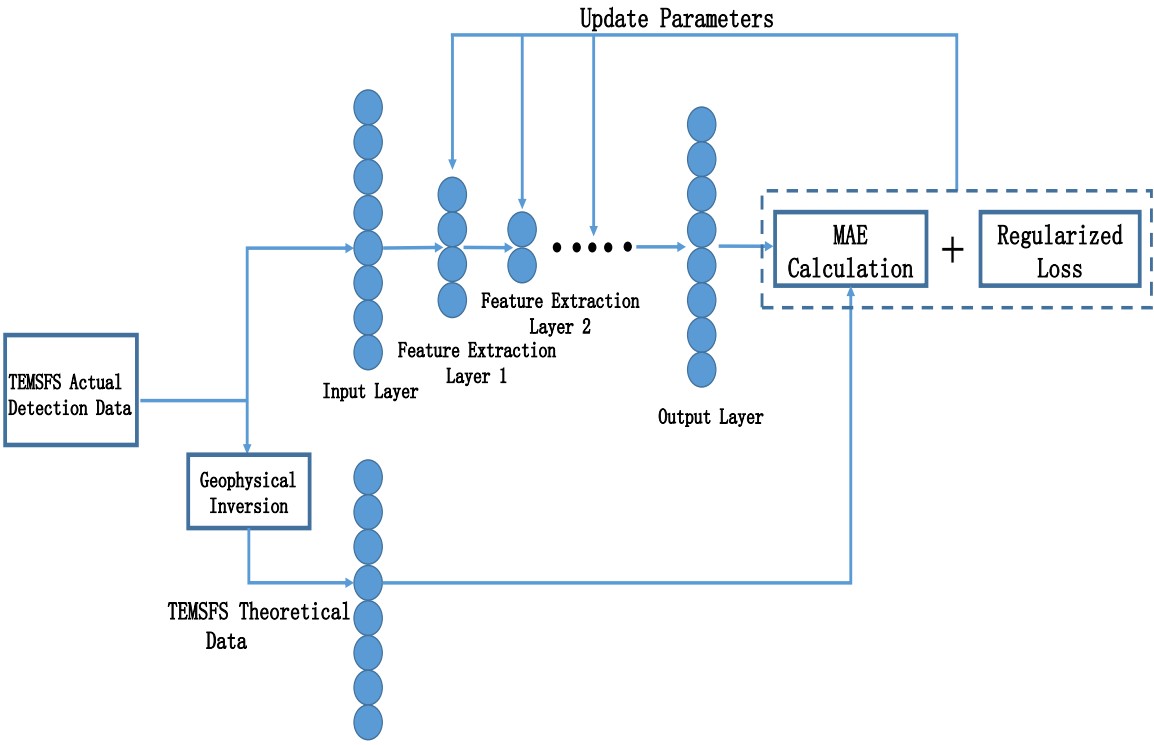

**Figure 1.** The flow chart of model. Total loss is sum of MAE calculation and regularization loss. MAE calculation is difference of theoretical signal and output. Regularization loss is calculated by L2. AEs are trained one by one and fine-tuning is used finally.

this small-scale input in this paper), and the more features extraction layers will inevitably lead to over-fitting. Moreover, the reconstruction effect can be affected by the number of features extraction layer nodes. If SFSDSA model has too few nodes, the characteristics of the data can not be learned well. However, if the number of features extraction layer nodes are too large, the designed lossy compression noise reduction can not be achieved well and the learning burden is increased.

5    Therefore, based on the aforementioned questions, we design the SFSDSA model(Figure 1), and the number of nodes in the latter features extraction layer is half the number of nodes in the previous features extraction layer, until finally reconstructed back to the original dimension. SFSDSA model is a layer-by-layer features extraction, which can be regarded as the process of stacking AE. Low dimensions are represented by the high-dimensional data features, which can learn the input features.At the same time, since the reconstruction loss is the loss of the output related with the clean signal, it can also be said that the input

10   signal can be regarded as a clean signal based on the noise, the training measure of DAE model increases the robustness of the model and reconstructs the lossy signal, and mapping the signal point to its high probability location can be viewed as a noise reduction process.

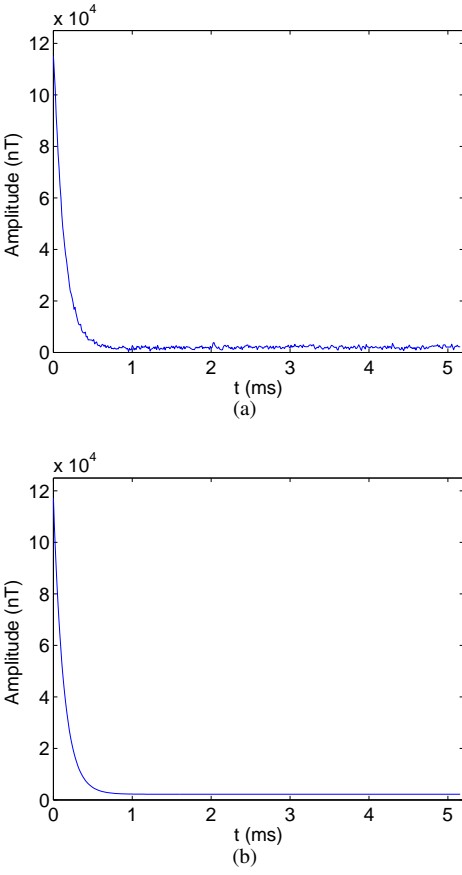

**Figure 2.** (a)Actual signal curves. (b) Inversion of theoretical signal curves.

**Table 1.** The training cost of combination of learning rate and regularization rate. The value represents the MAE of the first fifty data points. According to the experience, about the first fifty data points have better effect for extracting time-domain order waveforms

| learning rate \ regularization rate | 0.05 | 0.1 | 0.15 | 0.2 |
|---|---|---|---|---|
| 0.1 | 61515.3 | 12670.3 | 14448.9 | 11112.1 |
| 0.01 | 1735.2 | 1918.1 | 2126.6 | 1825.7 |
| 0.001 | 1526.6 | 1669.5 | **1377.3** | 1780.6 |
| 0.0001 | 1493.2 | 1678.1 | 1392.3 | 1955.5 |

In the training experiment, we collected 2400 periods of transient electromagnetic method secondary field signals from the same collection location, and selected 434 data points in per period. Meanwhile, 100 periods of signals are randomly acquired as a test and validation set for the improving the robustness of the model. We use Google's deep learning framework–Tensorflow, which is used to build the SFSDSA model. The parameter settings for the model are as follows: batch-size = 8,

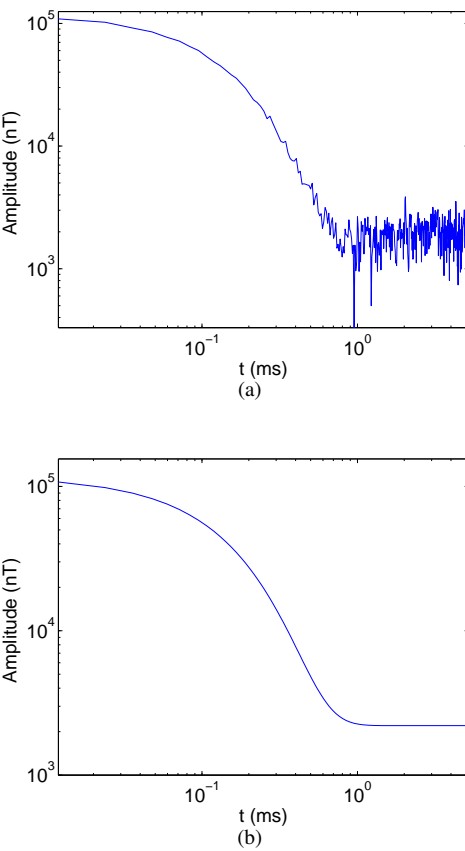

**Figure 3.** (a)Actual detection signal. (b) Inversion of theoretical detection signal.

epochs = 2. We do a grid search and get the good parameter combination of learning rate and regularization rate as shown in table 1(learning rate=0.001 and regularization rate=0.15).

We analyzed and compared the selection of the two loss functions of MAE and MSE in experiments as shown in figure 4. Meanwhile, according to the previous work and the SFS denoising task of transient electromagnetic method, we think that MAE is a better choice. On the one hand, our task is to map the outliers affected by noise to the vicinity of the theoretical signal point, in other words, model should ignore the outliers affected by noise to make it more consistent with the distribution of the overall signal. We know that MAE is quite resistant to outliers(Shukla, 2015). On the other hand, the squared-error is going to be huge for outliers, which tries to adjust the model according to these outliers on the expense of other good-points(Shukla, 2015). For signal that are subject to noise interference in the secondary field of transient electromagnetic method, we don't want to over-fitting outliers that are disturbed by noise, but we want to treat them as noise interfered data.The evaluation index

is the mean absolute error(MAE) of output reconstruction data and clean input data.The smaller the MAE, the closer the output reconstruction data is to the theoretical data.The model also performs better in noise reduction.

$$MAE(x,y) = \frac{1}{m}\sum_{i=1}^{m}|h(x)^{(i)} - y^{(i)}| \tag{9}$$

where x denotes the noise interference data, m denotes the number of sampling points, h denotes the model and y denotes theoretical data.

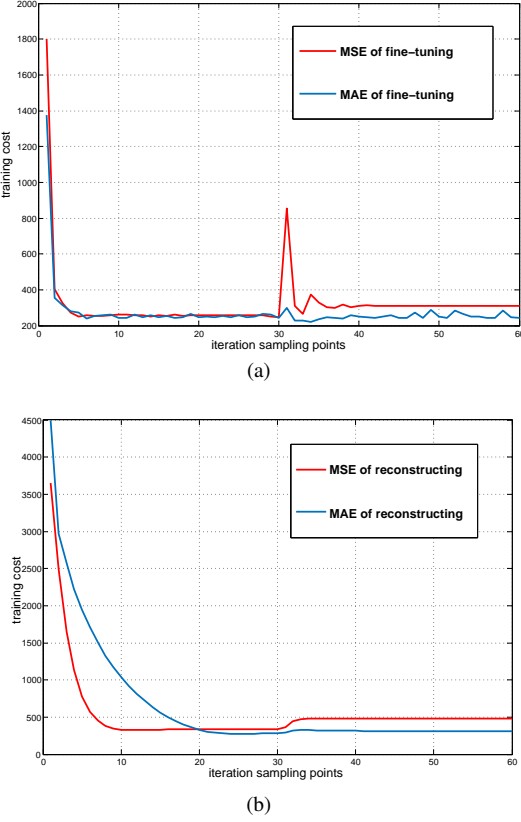

(a)

(b)

**Figure 4.** the training cost comparison of MAE and MSE, the left is the training cost of fine-tuning, and the right is the training cost of reconstructing.

In the previous experiments, we set hyper-parameters (batch-size=8, learning-rate=0.1, regularization-rate=0, epochs=20) based on experience but we initially take the measure of a small number of epochs (epochs=2) according to experiment. We added the experiment as shown in Figure 5 to support our standpoint. The model oscillates quickly and converges. Training with fewer epochs can avoid useless training and over-fitting, maintaining the distribution characteristics of the signal itself. As shown in Figure 6, the reconstruction error oscillates and converges as the training progresses. This phenomenon is similar

to the tail of the actual signal.We try stoping training when the convergence occurs, the idea similar to early-stopping makes the model more robust(Caruana et al., 2000).

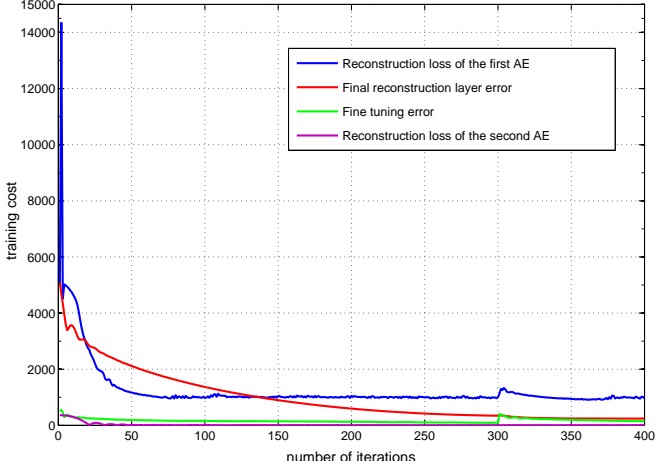

**Figure 5.** training cost of each process.

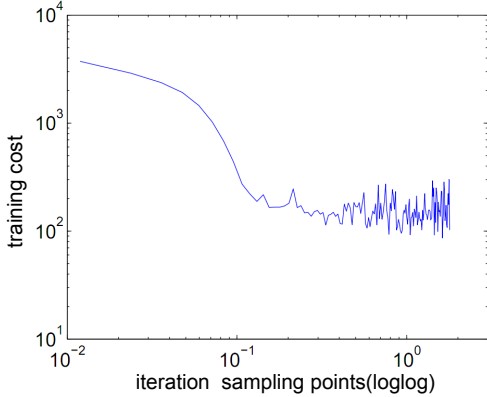

**Figure 6.** Iteration sampling points.

By analyzing Figure 7, the relationship between MAE and the number of hidden layers, we found that the result of stacking two AEs have good effect. We guess that the size of the AE hidden layer is too small after multiple stacking(for instance, the 5   4th AE only has 27 nodes because the size of latter AE is half of the previous AE in order to extract the better feature), and the representation of signal characteristics are not complete resulting in large reconstruction costs. If we want to get a better

result, more iterations may be used but this tends to cause over-fitting. Meanwhile, we found that the reconstruction loss of the second AE is already very small shown in Figure 5. So it is not necessary to stack more AEs.

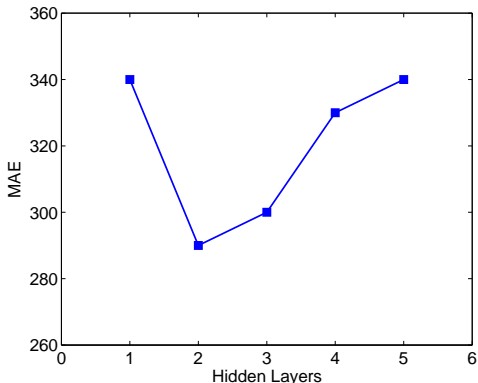

**Figure 7.** SFSDSA hidden layer number and MAE values.

Small-scale deep learning model, and the training times can be less. By analyzing Figure 2(a), we found that because the amplitude of the tail of the actual signal is small, and the influence of the noise is significant, so the tail of the signal oscillates violently.Meanwhile, after the feature extraction and noise reduction to a certain extent, the noise interference can not be completely removed, and the reconstruction can not completely present the clean signal, and it is only possible to map the signal points as high probability points as possible to reduce reconstruction loss.

### 4.1 Training results

After several experiments, the MAE of actual signals fell from 534.5 to about 215.Compared with the secondary field actual signals and signals denoised by SFSDSA model, the noise reduction effect of SFSDSA is obvious in Figure 8.

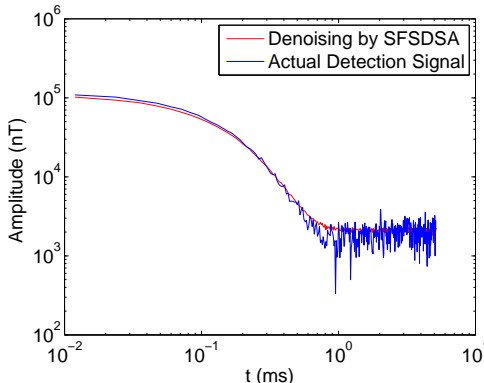

**Figure 8.** Actual secondary field data after SFSDSA model noise reduction

The 35th to 55th points are selected for specific analysis in Figure 7 . Through noise reduction of training good SFSDSA model, the singular points(large amplitude deviation from theoretical signal) affected by the noise are mapped to the high probability positions(e.g. no.38 point, no.51 point).This process is the process of damage reconstruction that the DAE model has verified.At the same time, our stacked AEs model also keeps on extracting the features, and the singular points are restored to the corresponding points according to the characteristics of the data.The whole process realizes the noise reduction of the secondary field actual signal based on the secondary field theoretical signal, and the model maps the singular points to locations where there is a high probability of occurrence, which is also similar to the most estimative method based on observations and model predictions by Kalman filtering.

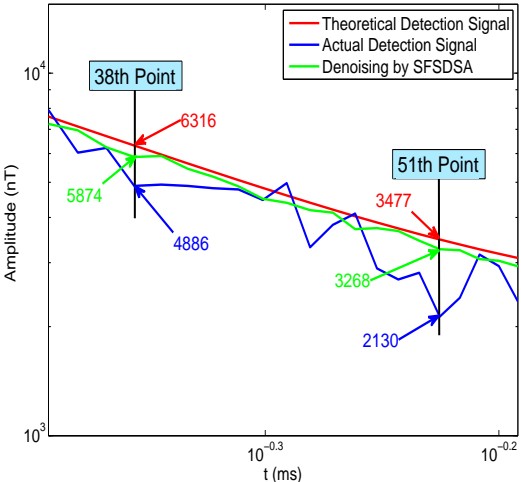

**Figure 9.** Specific points analysis.

## 5   Comparison with traditional noise reduction methods

We also conducted wavelet transform. PCA method and Kalman filter experiments, in which the number of layers of the wavelet transform is three, DWT () and construction function IDWT () is called in Matlab.By using the PCA method, we do the experiment to verify the effect of noise reduction. But the process of programming is more complicated using mathematical derivation, so we use scikit-learn library to realize noise reduction.Kalman filtering is implemented in Python, where the system noise Q is set to 1e-4 and the measurement noise R is set to 1e-3. The Figure 10 shows the absolute error distribution for that method.We can find from the figure model of noise reduction based on SFSDSA of secondary field data, SFSDSA is better than kalman filter, wavelet transform and PCA method. At the same time, as the kalman filter is a linear filter, its noise reduction effect is so poor in this paper. Moreover, the underlying structure is not easy to modify resulting in scikit-learn library is unable

**Table 2.** Comparison of MAE models

| Model Name | Parameter Setting | MAE |
|---|---|---|
| SFSDSA+SELU+REGULARAZTION+ADAM | Learning_rate_base:0.00103 | 150.36 |
| SFSDSA+RELU+REGULARAZTION+ADAM | Learning_rate_base:0.00103 | 1500.20 |
| SFSDSA+SELU+ADAM | Learning_rate_base:0.00103 | 164.30 |
| SFSDSA+SELU+REGULARAZTION | Learning_rate_base:0.00103 | 5112.30 |
| Wave Transform | Three layers of wavelet transform | 451.20 |
| Kalman Filter | Q=e-4 R=e-3 | 503.20 |

to adjust parameters adaptively based on signal characteristics. After the filtering test, and then the MAE corresponding to the calculation of the theoretical data, it can be seen that the effect of pca filtering is lower than SFSDSA.

At the same time, we compared the optimization results of various models using the traditional method with those of the SFSDSA model, as shown in table 2.

Figure 11 is the diagram of the mine where the exploration experiment was conducted. The red thick curve is the actual mine vein curve. A data collection survey line, which is the southwest-northeast pink curve shown in the figure, is designed with seven points marked as number 1 to 7 along it, and the distance between each point is 50 meters.

In the data analysis, we analyzed the first 50 points in the second field which collected in actual mine. The early signal of the secondary field is stronger than later, and it is not easy be disturbed by the noises. So in the Figure 12, we take out the later 21 points in each collection point, which is used for further analysis. Figure 12 (a) is extracted time-domain order waveforms formed by the actual data acquired at the seven collection points at the same time.Figure 12 (b) extracted time-domain order waveforms formed by the data denoised by SFSDSA model. By comparing the two images in Figure 12, it can be clearly seen that the curves in Figure 12(a) have obvious intersections, and the intersections in Figure 12(b) can't be seen almost.In transient electromagnetic method, the intersected curve can't indicates the deeper underground geological information.It can be explained that the curve after the de-noising model can reflect the deep geological information.

## 6   Conclusions

Based on the transient electromagnetic method, the deep-seated information is reflected in the late-stage of the second field signal when deep-level surveys are conducted. but the late-stage signals are very weak and easily contaminated by noise. Therefore, we use the measured data for modeling to obtain the theoretical model, which will perform noise reduction based on the geological features represented by the previous training data-set. Meanwhile, it is necessary to analyze the known geological features carefully and apply the model according to the actual geological conditions before using our method. And this method has good generalization for different collection points of the same geological feature area. By introducing the deep learning algorithm integrated with the characteristics of the secondary field data,SFSDSA can map the contaminated signal to

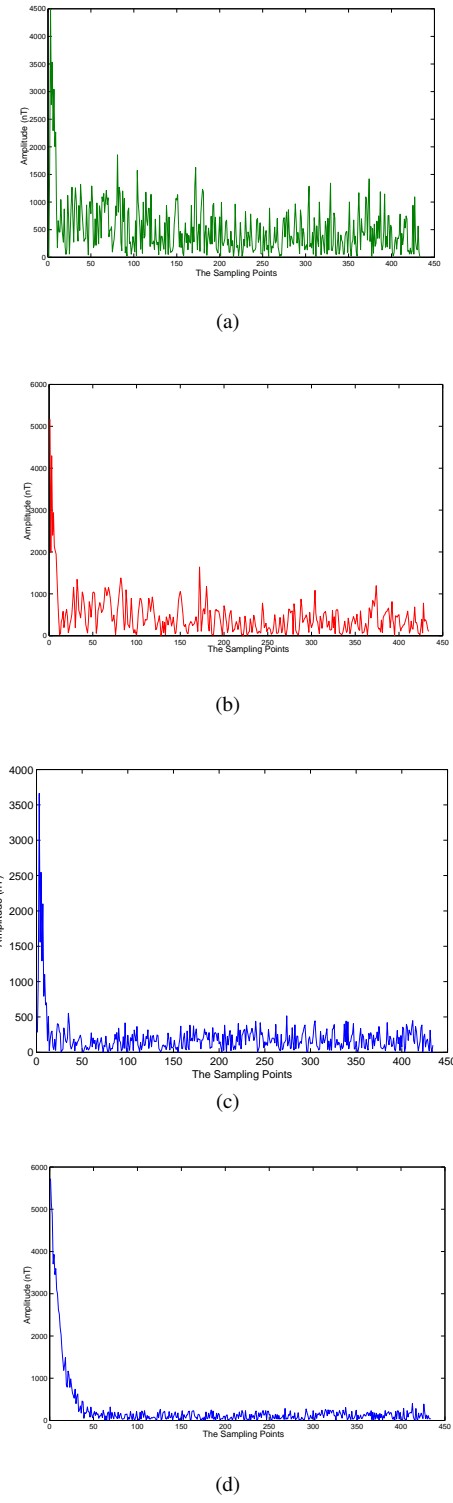

**Figure 10.** (a)Kalman filter. (b)Wavelet transform filter. (c)PCA filter. (d)SFSDSA denoising.

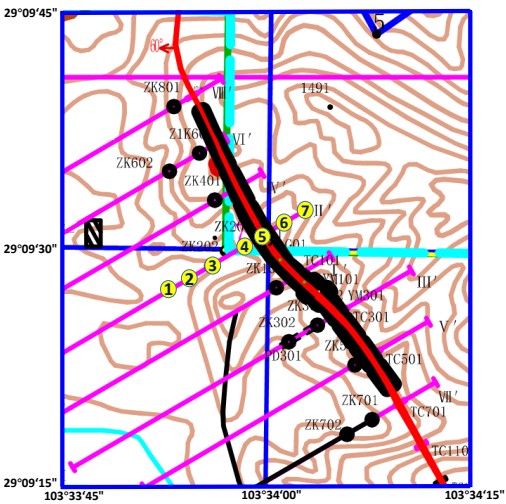

**Figure 11.** The geographic distribution of the collection points(1th to 7th).

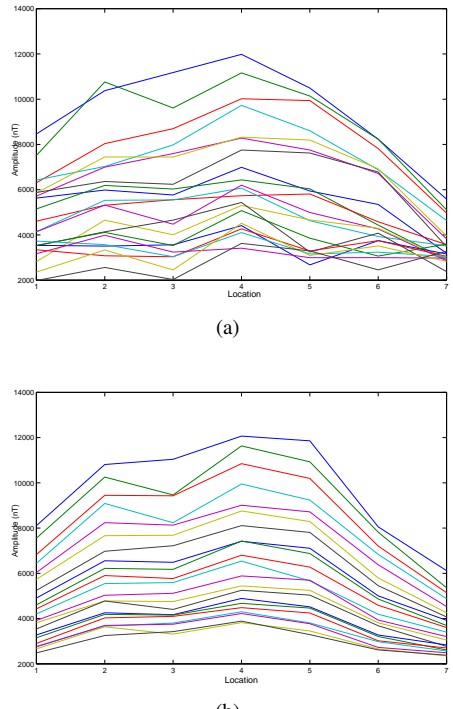

(a)

(b)

**Figure 12.** (a)The Original 30th to 50th points from seven actual detecting locations. (b)The denoising 30th to 50th points from seven actual detecting locations.

a high probability position. By comparing several filtering algorithms by using same sample data , the SFSDSA method has better performanc and the denoising signal is conducive to further improve the effective detection depth.

*Code availability.* The codes are available by email request

*Data availability.* The data sets are available by email request

5  *Code and data availability.*

*Sample availability.*

*Video supplement.*

**Appendix A**

**A1**

10  *Author contributions.* FQ proposed and designed the main idea in this paper. KC completed the main program and designed main algorithm, XB instructed all the authors. Hui gave many meaningful suggestions, other authors participated in the experiments and software development.

*Competing interests.* The authors declare that they have no conflict of interest.

*Disclaimer.*

*Acknowledgements.* This paper is supported by National Key R&D Program of China (NO.2018YFC0603300). The authors thank three anonymous referees for their careful and professional suggestions to improve this paper and the authors thank Sunyuan Qiang who completed a part of PCA programming at stage of solving the question for 3th referee.

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
