# Peer review of "A denoising stacked autoencoders for transient electromagnetic signal denoising"

_Nonlinear Processes in Geophysics, 2018_

## Referee Comment (RC1) · Anonymous Referee #1 · 19 Nov 2018

This paper is about the deep learning of geophysics, especially the application of stacked denoising auto-encoder in transient electromagnetic method. This is an innovative application point with innovation. After I read it, the following questions should be addressed carefullyïijŽ 1ãĂĄThe sample selection for learning is not clearly described in the paper. Please add the sample selection instructions, and specify the source of the samples and the selection principles. 2ãĂĄPlease describe the parameters of the experimental platform, the hyper-parameter indexes in the model, and the code of the main model module. 3ãĂĄPlease add more description of the specific parameters and experimental details of kalman filter and wavelet transform. 4ãĂĄThe related work section over introduces the encoder related literature. Please elaborate the literature of transient electromagnetic signal noise reduction and signal filtering. 5ãĂĄThe loss

in figure 1 is calculated as two regularized losses. What does that mean? It should be explain detail. 6ãĂĄPlease explain the relationship between figure 5 and the corresponding explanatory text. And whether the coordinates in Figure 5 is correctïij§

---

## Referee Comment (RC2) · Anonymous Referee #2 · 3 Dec 2018

Authors presented a deep learning method to suppress noise in the transient electromagnetic method. It's an interesting and well-written paper. Please find below my comments.

1. How exactly you are planning to train the network on realistic geophysical problems. 2. Is this method can be generalized in the sense that the training on one data can be used on different datasets? 3. Any comments on using supervised learning since that seems work better than the unsupervised learning? 4. If noise is not random as shown in the examples, will this method still work? 5. What the runtime cost of the proposed method compared to other denoising methods?
* * *
2018-39, 2018.

---

## Referee Comment (RC3) · Anonymous Referee #3 · 5 Dec 2018

**A denoising stacked autoencoders for transient electromagnetic signal denoising**

The paper presents a denoising method based on autoencoders, which learn "automatically" a lossy compression based on the input data with an unsupervised manner. In this paper, the authors apply autoencoders to TEMSFS data in order to reduce the potential existing noise by compressing and then reconstructing it. In this paper, the reconstructed signal is compared to the theoretical one.

The research carried out in this paper is relevant by means to the application to geophysical signals, which has not been applied before (according to the authors). However, in my opinion the paper needs major revision.

Important comments:
My field is machine learning and data science and not geophysics.
- The paper aims to denoise a signal with autoencoders (unsupervised manner). However, the authors did this by putting a theoretical signal as output. This is not an unsupervised manner to proceed. Why did the authors put an output? Is it a traditional way to proceed in geophysics?
- Page 2 line 3: can you please explain little bit why PCA is cumbersome and what could be the effect on the signal used as case study?
- Page 3, in which the SELU activation function and the ADAM optimisation algorithm are introduced, a justification of choice is needed.
- Page 3 line 24: "SELU activation function is utilized to prevent too many of depth": please put a reference for that? Same page line 12, authors said: SELU and ADAM optimisation algorithm are used to solve the problem of over-fitting. How? Need references for this point or a good justification.
- Please add other criteria in addition to the MAE.
- The data splitting need more explanations. The experimental case study needs also some explanation with some exploratory analysis.
- For choosing only 2 hidden layers, did you take into account the other hyper parameters. I suggest a grid search, which is possible to do using TensorFlow library or Keras in Python.
- For the comparison with traditional methods, please add PCA.
- Explain how the traditional methods were applied (mother wavelet …).

Some remarks:

- Put more explanation on the caption of figure 1 if possible.
- Equation 9: put bracket.  In addition, explain it little bit (m, X, h …) if possible.
- Page 6 line 10: the authors used Tensorflow, please put a figure of the architecture of the used model.
- Since the journal is open source, think to put your code on an open source platform (e.g. GitHub …)

---

## Author Comment (AC1) · 14 Jan 2019

Author's Response for comments of review 1

Dear Anonymous Referee: We very much appreciate the overall positive attitude of the referee to our manuscript and thank you for your time and very useful comments! We give below responses to some of these. And we also give a marked-up version and a revised version.

1.The sample selection for learning is not clearly described in the paper. Please add the sample selection instructions, and specify the source of the samples and the selection principles. Reply: In the training experiment, we collected 2400 periods of transient electromagnetic method secondary field signals from the same collection location, and

selected 434 data points in per period. Meanwhile, 100 periods of signals are randomly acquired as a test and validation set for the improving the robustness of the model.

2.Please describe the parameters of the experimental platform, the hyper-parameter indexes in the model, and the code of the main model module. Reply: Thank the reviewer for this comment about hyper-parameters, so we accept your suggestion to add more experiment details about learning rate, batch-size and so on. More details can be find in marked-up manuscript. Meanwhile, we upload the code of the main model module on github: https://github.com/tonyckc/SFSDSA.

3.Please add more description of the specific parameters and experimental details of kalman filter and wavelet transform. Reply: This question was described in detail in the fifth part of the original manuscript.

4.The related work section over introduces the encoder related literature. Please elaborate the literature of transient electromagnetic signal noise reduction and signal filtering. Reply: Yes, we introduce fewer related works about transient electromagnetic method signal noise reduction and signal filtering. Therefore, we describe more related works about TEM denoising using other Kalman filter and Wavelet transform method, and then adding more references. More details can be find in marked-up manuscript.

5.The loss in figure 1 is calculated as two regularized losses. What does that mean? It should be explain detail. ReplyïijˇZ We are very sorry that make you think that we calculate two regularized losses because of our error figure. In fact, in Page 4 Line 14 we described that the theoretical signal is used to get the model loss with the output using the activation function. Meanwhile, we have replaced Figure 1 to a clear version.

6.Please explain the relationship between figure 5 and the corresponding explanatory text. And whether the coordinates in Figure 5 is correctïij§ ReplyïijˇZ In the part 4, we narrated about the figure 5 detailedly. Small-scale deep learning model, and the training times can be less. As is shown in Figure 5, we set each batch to 8 and every 10 training steps output MAE value as a iteration sampling point. We found that the

MAE values near the 150th sampling point which will start oscillate. Both coordinates of the figure is logarithmic, the two coordinates have been modified. That can be seen in modified manuscript .

Please also note the supplement to this comment:
https://www.nonlin-processes-geophys-discuss.net/npg-2018-39/npg-2018-39-AC1-supplement.pdf

[Figure]

**Fig. 1.**

[Figure]

**Fig. 2.**

**Supplement:**

[revised manuscript text omitted]

---

## Author Comment (AC2) · 15 Jan 2019

Dear Anonymous Referee:

We very much appreciate the overall positive attitude of the referee to our manuscript and thank you for your time and very useful comments! We give below a first response to some of these. Meanwhile, according to your comments, we revised this manuscript. All of the changes were made in a marked-up manuscript version and a clear revised version.

1. Comment from Reviewer: How exactly you are planning to train the network on realistic geophysical problems?

Reply: We agree with your comment. We explained how to train the network on real-

istic geophysical problems in the section 4 (Experiment and Analysis). However, we found that the explanation of this process was not clear after we carefully read the fourth part of the manuscript again. We briefly described the process: As described in the third part (Mathematical Derivation of SFSDSA), we can obtain an actual detection signal sample and a theoretical signal sample, and then we build a model for training. Meanwhile, Figure 1 shows the network structure and training process in a more vivid way. For realistic geophysical problems such as transient electromagnetic method secondary field signals, we carried out experiments in the fourth part according to the process proposed in the third part .We collect the actual detection signal of the secondary field, the dimensions are 1 * 434 (this dimension can reflect the attenuation process of transient electromagnetic method secondary field signal), the inversion theory signal and the actual detection signal of the secondary field has the same dimensions. Finally, we used two samples for training according to training process of the Figure 1(we added more details, such as training platform, hyper-parameters etc.). Meanwhile, inversion theory signals play a semi-supervisory role in the model. In the end, SFSDSA can map the signal points of the noise interference to the high probability points with clean signal as reference according to the deep characteristics of the signal, so as to realize the signal noise and reduce noise interference.

2. Comment from Reviewer: Is this method can be generalized in the sense that the training on one data can be used on different datasets?

Reply: Yes, this method has a good generalization in a certain sense. Our method has good generalization for different collection points of the same geological feature area. As shown in Figure 9, we use the same model for 7 collection points. However, if the acquisition areas of the two data have large differences in geological features, this will inevitably lead to different deep features of the forward and inversion signals that cause the secondary field. The model will perform noise reduction based on the geological features represented by the previous training dataset. Therefore, it is necessary to analyze the known geological features more carefully and apply the model according

to the actual geological conditions before using our method. At the same time, this view is consistent with machine learning theory(Neyshabur et al,.2017). If the model will be well generalized, it must be built to varying degrees of similarity problems. If we do not analyze the principle of the problem and ignore the huge differences in features, it is unrealistic to try to achieve a high degree of generalization. According to this comment, we added this view to the part of conclusion the marked-up manuscript.

3. Comment from Reviewer: Any comments on using supervised learning since that seems work better than the unsupervised learningïij§

Reply: Recently, we have noticed that supervised learning performs well in classification problems such as image recognition and semantic understanding(He et al.,2016, Long et al.,2014). At the same time, unsupervised learning also has a good performance in clustering and association problems (Klampanos et al., 2018.), and the goal of unsupervised learning is usually to extract the distribution characteristics of the data in order to understand the deep features of the data (Becker et al.,1996, Liu et al., 2015). Both supervised learning and unsupervised learning have their own well-behaved areas, so we need to choose different learning styles and models for different problems. For the noise suppression problem of the secondary field signal in transient electromagnetic method, our goal is to extract the deep features of the secondary field signal, and map the data points affected by noise to the estimated high probability points according to their own signal features. We also found that the purpose of extracting the distribution characteristics of the secondary field signal data is similar to that of unsupervised learning. Meanwhile, unsupervised learning models are widely used in different signal noise reduction problems, some of which perform well such as gravitational waves, power transmission equipment status signals, etc. According to this comment, we added this view to the part of relate work in marked-up manuscript.

4. Comment from Reviewer: If noise is not random as shown in the examples, will this method still work?

Reply: Yes, Our model can extract features of the secondary field signal, so as the signal points of noise interference are mapped to the estimated high probability points according to their own signal characteristics. From a very natural point of view, noise can be seen as an interference whether it is random or not. At the same time, the deep learning model has a good generalization feature to support our point of view (Neyshabur et al., 2017), we also added measures to improve generalization in SFS-DSA such as regularization (Nowlan and Hinton., 1992), so our method has a better performance in actual tests such as the results of Figure 8 and Figure 10. Therefore, this method is still work in a certain sense if the noise is not random.

5. Comment from Reviewer: What the runtime cost of the proposed method compared to other denoising methods?

Reply: Our runtime cost are less at the end of training compared to other denoising methods such as wavelet transform. We can use the data with noise to achieve end-to-end denoising (as described in the process of Figure 1) using the trained model, without having to spend a lot of time to adjust the wavelet threshold and wavelet base like wavelet transform. For small sample data sets, the time consumption difference between SFSDSA and other denoising methods is small, but when the number of data samples reach a certain quantity, the model has a higher advantage in time consumption after training.

We appreciate all the comments, which we will use to improve the manuscript.

References He K , Zhang X , Ren S , et al. Deep Residual Learning for Image Recognition[J]. IEEE Conference on Computer Vision and Pattern Recognition (CVPR),2016, DOI:10.1109/CVPR.2016.90

Long J., Shelhamer E., Darrell T.. Fully Convolutional Networks for Semantic Segmentation[J]. IEEE Transactions on Pattern Analysis & Machine Intelligence, 39(4):640-651, 2014.

Klampanos I A , Davvetas A , Andronopoulos S , et al.: Autoencoder-driven weather clustering for source estimation during nuclear events, Environmental Modelling & Software, 102:84-93,.2018.

Becker S , Plumbley M.: Unsupervised neural network learning procedures for feature extraction and classification. Applied Intelligence: The International Journal of Artificial, Intelligence, Neural Networks, and Complex Problem-Solving Technologies,6(3):185-203. 1996.

Liu J H , Zheng W Q , Zou Y X . A Robust Acoustic Feature Extraction Approach Based on Stacked Denoising Autoencoder[C]// IEEE International Conference on Multimedia Big Data. IEEE Computer Society, 2015.

Neyshabur B , Bhojanapalli S , Mcallester D , et al. Exploring Generalization in Deep Learning., 2017.

S. J. Nowlan and G. E. Hinton. Simplifying neural networks by soft weight-sharing. Neural Computation, 4(4), 1992.

Please also note the supplement to this comment:
https://www.nonlin-processes-geophys-discuss.net/npg-2018-39/npg-2018-39-AC2-supplement.pdf

**Supplement:**

[revised manuscript text omitted]

---

## Author Comment (AC3) · 15 Jan 2019

Author's Response for comments of review 3

Dear Anonymous Referee: Thank for you positive comments to our manuscript! We give below responses to some of these. Meanwhile, according to your comments, we revised this manuscript overhaul. All of the changes were made in the supplement files, which are a marked-up version and a revised version.

1. Comment from Reviewer: "The paper aims to denoise a signal with autoencoders (unsupervised manner). However, the authors did this by putting a theoretical signal as output. This is not an unsupervised manner to proceed. Why did the authors put an output? Is it a traditional way to proceed in geophysics?" Reply: We are very sorry

that make you think that we put a theoretical signal as output because of our unclear expression in Figure 1(in original manuscript). In fact, in Page 4 Line 14, we described that the theoretical signal is used to get the model loss with the output using the loss function, to realize back propagation. Meanwhile, we have replaced Figure 1 with a clear version. For another question, naturally, it is not a traditional way to proceed in geophysics. We have modify manuscript according to this comment.

2. Comment from Reviewer: "Page 2 line 3: can you please explain little bit why PCA is cumbersome and what could be the effect on the signal used as case study? Reply: According to the references (Wu et al., 2014), the process of PCA can be divided into 5 steps. (1) Normalize the obtained data (2) Calculate the covariance matrix for obtaining multidimensional data (3) Decompose the covariance matrix to obtain the eigenvalue matrix and eigenvector (4) Obtain the corresponding main components after dimensionality reduction according to the PCA calculation method (5) Selecting the representative principal components by the trend comparison method and the L-curve method, and performing reconstruction to obtain the denoised secondary field signal waveform. By using the PCA method, we do the experiment to verify the effect of noise reduction. But the process of programming is more complicated using mathematical derivation, so we use scikit-learn library to realize noise reduction. However, the underlying structure is not easy to modify resulting in scikit-learn library is unable to adjust parameters adaptively based on signal characteristics. Meanwhile, we found that the filtering effect is not ideal. More details can be find in revised manuscript.

3. Comment from Reviewer: "Page 3, in which the SELU activation function and the ADAM optimization algorithm are introduced, a justification of choice is needed." Reply: The problem of too many nodes dying is a general disadvantage for RELU activation function and improved RELU activation functions like leaky RELU all consistently outperform the RELU in some tasks (Xu et al.2015). Therefore, it is necessary to apply the improved RELU function to reduce the impact of the shortcomings of the RELU function. We choose the SELU that have the preponderances of overcoming vanishing and exploding gradient problems in a sense and the best preforming in full connection networks (Klambauer et al., 2014). Adam algorithm have the advantages of calculating different adaptive learning rates for different parameters and requiring little memory(Kingma et sl.,2014). Through Table 1(in original manuscript), we find that the combination of models using SELU is better than the combination of models using RELU in the MAE indicator. Similarly, we find that the combination of models using ADAM optimization algorithm outperform compared with not using ADAM in the MAE indicator. More details can be find in revised manuscript.

4. Comment from Reviewer: "Page 3 line 24: "SELU activation function is utilized to prevent too many of depth": please put a reference for that? Same page line 12, authors said: SELU and ADAM optimization algorithm are used to solve the problem of over-fitting. How? Need references for this point or a good justification." Reply: we are very sorry that the sentence of "SELU activation function is utilized to prevent too many of depth" has a spelling mistake (the word 'depth' should be replaced to 'death') to lead to an unclear and incorrect description. In fact, this sentence wants to express that SELU is utilized to reduce the impact of too many dying nodes problem(Xu et al.2015, Klambauer et al., 2014). For the second question in page 3 line 12, our description of function of SELU and Adam is unclear because of the poor grammar. In fact, we chose Adam algorithm, which have the advantages of calculating different adaptive learning rates for different parameters and requiring little memory(Kingma et sl.,2014). And SELU have the preponderances of overcoming vanishing and exploding gradient problems in a sense and the best preforming in full connection networks (Klambauer et al., 2014). We changed the description of the part to a correct expression. More details can be find in revised manuscript.

5. Comment from Reviewer:"Please add other criteria in addition to the MAE" Reply: In fact, we analyzed and compared the selection of the two loss functions of MAE and MSE in the previous experiments as shown in figure 1. Meanwhile, according to the previous work and the secondary field signal denoising task of transient electromagnetic method, we think that MAE is a better choice. First, our task is to map the outliers affected by noise to the vicinity of the theoretical signal point, in other words, model should ignore the outliers affected by noise to make it more consistent with the distribution of the overall signal. We know that MAE is quite resistant to outliers(Rishabh, 2015), so we choose it. Second, the squared-error is going to be huge for outliers, which tries to adjust the model according to these outliers on the expense of other good-points(Rishabh, 2015). For signal that are subject to noise interference in the secondary field of transient electromagnetic method, we don't want to over-fitting outliers that are disturbed by noise, but we want to treat them as noise interfered data. Finally, observing the secondary field signal of transient electromagnetic method, we found that the amplitude of the early track data points is very large, but the amplitude of the late track data is small, and the squared-error will inevitably give the early points of the abnormal points more weight to result in Ignoring the difference in late-channel data, this is very unfair. This question may lead to inaccurate model and late-channel signals will be ignored. We have modify manuscript according to this comment.

6. Comment from Reviewer:"The data splitting need more explanations. The experimental case study needs also some explanation with some exploratory analysis" Reply: In the previous experiment, we randomly collected 2400 periods of transient electromagnetic method secondary field signals from the same collection location and we collected 434 signal points per period. Meanwhile, 100 periods of signals are randomly acquired as a test and validation set. In the meantime, we accept the second suggestion to do some explanation with some exploratory analysis in reply 7 and we update the manuscript for adding more details. We have modify manuscript according to this comment.

7. Comment from Reviewer: "For choosing only 2 hidden layers, did you take into account the other hyper parameters. I suggest a grid search, which is possible to do using TensorFlow library or Keras in Python". Reply: Thank the reviewer for this precious and professional comment about hyper parameters, and we're so sorry that

this paper doesn't list some important hyper parameters such as learning rate, regular parameter and so on completely. We have added those key hyper-parameters in marked-up manuscript. In the previous experiments, we set hyper-parameters (batch-size=8, learning-rate=0.1, regularization-rate=0, epochs=20) based on experience but we initially take the measure of a small number of epochs (epochs=2) according to experiment. We added the experiment as shown in Figure 2 to support our standpoint. The model oscillates quickly and converges. Training with fewer epochs can avoid useless training and over-fitting, maintaining the distribution characteristics of the signal itself. As shown in Figure 5(in original manuscript), the reconstruction error oscillates and converges as the training progresses. This phenomenon is similar to the tail of the actual signal. We try stopping training when the convergence occurs, the idea similar to early-stopping makes the model more robust(Caruana ,2000). At the same time, we got the result of stacking two AEs with good effect as shown in Figure 4(in original manuscript). We guess that the size of the AE hidden layer is too small after multiple stacks (for instance, the 4th AE only has 27 nodes because the size of latter AE is half of the previous AE in order to extract the better feature), and the representation of signal characteristics are not complete resulting in large reconstruction costs. If we want to get a better result, more iterations may be used but this tends to cause over-fitting. Meanwhile, we found that the reconstruction loss of the second AE is already very small shown in Figure 2. And it is not necessary to stack more AEs. We accept the reviewer's suggestion to do a grid search, and we get the good parameter combination of learning rate and regularization rate as shown in table 1 in revised manuscript (learning rate=0.001 and regularization rate=0.15).

8. Comment from Reviewer:" For the comparison with traditional methods, please add PCA." ReplyïijŽWe have already added in the manuscript about the comparison of PCA algorithm in transient electromagnetic signal denoising. After the filtering test, and then the MAE corresponding to the calculation of the theoretical data, it can be seen that the effect of pca filtering is lower than SFSDSA. Please see the fifth part of the article for details.

9. Explain how the traditional methods were applied (mother wavelet ...). ReplyïijŽA denoising algorithm utilizing wavelet threshold method and exponential adaptive window width-fitting(Ji et al.,2016). An exponential fitting algorithm was used to achieve the attenuation curve for each window, and the data contaminated with non-fixed electromagnetic noise was replaced by their results. Another algorithm utilizes multi-resolution analysis via a stationary wavelet transform of the data(Li et al.,2017).The measured data are decomposed into detailed coefficients and approximated coefficients. Then, the logarithmic slope of measured data and a threshold are calculated to identify the noise in the detailed coefficients; the corresponding detailed coefficients are processed to reduce the noise. Finally, the undisturbed data are reconstructed using inverse stationary wavelet transform. The third method presents an exponential fitting-adaptive Kalman filter to remove mixed electromagnetic noises(Ji et al.,2017), while preserving the signal characteristics. It consists of an exponential fitting procedure and an adaptive scalar Kalman filter. The adaptive scalar Kalman uses the exponential fitting results in the weighting coefficients calculation. Another wavelet-based baseline drift correction method for grounded electrical source airborne transient electromagnetic signals(Wang et al.,2013), through simulations, this method can improve the signal-to-noise ratio. Simulation results show that the wavelet-based method outperforms the interpolation method. All above were added in manuscript at the part of Related work.

Response for some remarks: 1. Put more explanation on the caption of figure 1 if possible. Reply: We accept this suggestion to put more explanation on the caption of figure 1. More details can be find in revised manuscript. 2. Equation 9: put bracket. In addition, explain it little bit (m, X, h ...) if possible. Reply: We are so sorry that we miss bracket on the right of 'x', and the input value of MAE should revised to 'x' and 'y'. x denotes the noise interference data, m denotes the number of sampling points, h denotes the model and y denotes theoretical data. The revised formula can be find in marked-up manuscript.

3. Page 6 line 10: the authors used Tensorflow, please put a figure of the architecture of the used model. Reply: The figure is exporting from Tensorboard GRAPHS to show the architecture of used model. https://github.com/tonyckc/SFSDSA/blob/master/The%20model%20structure%20.png

4. Since the journal is open source, think to put your code on an open source platform (e.g. GitHub ...) Reply: Code can be find: https://github.com/tonyckc/SFSDSA.

We appreciate all the comments, which we will use to improve the manuscript.

References: Caruana R, Lawrence S, Giles L.,: Overfitting in neural nets: backpropagation, conjugate gradient, and early stopping, International Conference on Neural Information Processing Systems. MIT Press, 2000.

Ji, Y., Li, D., Yu, M.,Wang,Y.,Wu,Q.,Lin,J. :A de-noising algorithm based on wavelet threshold-exponential adaptive window width-fitting for ground electrical source airborne transient electromagnetic signal, Journal of Applied Geophysics., 128,1-7,DOI: 10.1016/j.jappgeo.2016.03.001,2016.

Ji, Y., Wu, Q., Wang, Y.,Lin,J.,Li,D.,Du,S.,Yu,S.,Guan,S.: Noise reduction of grounded electrical source airborne transient electromagnetic data using an exponential fitting-adaptive Kalman filter, Exploration Geophysics., DOI: 10.1071/EG16046 ,2017.

Klambauer, G., Unterthiner, T., Mayr, A.,Hochreiter, S.: Self-Normalizing Neural Networks, arXiv preprint,arXiv:1706.02515:1706.0251, 2014.

Li, D., Wang, Y., Lin, J.,Yu,S.,Ji,Y.: Electromagnetic noise reduction in grounded electrical-source airborne transient electromagnetic signal using a stationary wavelet-based denoising algorithm, Near Surface Geophysics.,1(-9),DOI: 10.3997/1873-0604.2017003 2017.

Rishabh.: L1 vs. L2 Loss function, http://rishy.github.io/ml/2015/07/28/l1-vs-l2-loss, 2015

Xu B, Wang N, Chen T, et al. Empirical evaluation of rectified activations in convolutional network[J]. arXiv preprint arXiv:1505.00853, 2015

Wang, Y., Ji, Y.J, Li,S.Y., Lin, J., Zhou, F.D., Yang, G.H.,: A wavelet-based baseline drift correction method for grounded electrical source airborne transient electromagnetic signals. Exploration Geophysics,44,229–237,2013.

Wu, Y., Lu, C. D., Du, X. Z., Yu, X.D.: A denoising method based on principal component analysis for airborne transient electromagnetic data, Computing Techniques for Geophysical and Geochemical Exploration.,36(2),170-176, DOI: 10.3969/j.issn.1001-1749.2014.02.08,2014.

Please also note the supplement to this comment:
https://www.nonlin-processes-geophys-discuss.net/npg-2018-39/npg-2018-39-AC3-supplement.pdf
* * *
[Figure]

**Fig. 1.** trainingcostofMAEandMSE

[Figure]

**Fig. 2.** trainingcostofeachprocess

[Figure]

**Fig. 3.** PCA

**Supplement:**

[revised manuscript text omitted]